evolution, developmental biology, palaeontology

mammals, mandible, developmental constraint, ecomorphology, convergence

**Author for correspondence:**
Anne-Claire Fabre
e-mail: fabreac@gmail.com

# Functional constraints during development limit jaw shape evolution in marsupials

Anne-Claire Fabre[1,3], Carys Dowling[1], Roberto Portela Miguez[1], Vincent Fernandez[2], Eve Noirault[1] and Anjali Goswami[1]

[1]Department of Life Sciences, and [2]Imaging and Analysis Centre, The Natural History Museum, London SW7 5DB, UK
[3]Palaeontological Institute and Museum, University of Zurich, Zurich, Switzerland

A-CF, 0000-0001-7310-1775; RPM, 0000-0003-3094-9949; VF, 0000-0002-8315-1458;
AG, 0000-0001-9465-810X

Differences in jaw function experienced through ontogeny can have striking consequences for evolutionary outcomes, as has been suggested for the major clades of mammals. By contrast to placentals, marsupial newborns have an accelerated development of the head and forelimbs, allowing them to crawl to the mother's teats to suckle within just a few weeks of conception. The different functional requirements that marsupial newborns experience in early postnatal development have been hypothesized to have constrained their morphological diversification relative to placentals. Here, we test whether marsupials have a lower ecomorphological diversity and rate of evolution in comparison with placentals, focusing specifically on their jaws. To do so, a geometric morphometric approach was used to characterize jaw shape for 151 living and extinct species of mammals spanning a wide phylogenetic, developmental and functional diversity. Our results demonstrate that jaw shape is significantly influenced by both reproductive mode and diet, with substantial ecomorphological convergence between metatherians and eutherians. However, metatherians have markedly lower disparity and rate of mandible shape evolution than observed for eutherians. Thus, despite their ecomorphological diversity and numerous convergences with eutherians, the evolution of the jaw in metatherians appears to be strongly constrained by their specialized reproductive biology.

## 1. Introduction

Shifts in development and life history can have a profound impact on the evolutionary trajectories of lineages. For example, changes in developmental timing of different body parts inherited from a common ancestor (known as heterochrony [1,2]) are known to have facilitated major evolutionary transitions in several clades (e.g. between dinosaurs and birds [3] and at each transition in the developmental strategy in salamanders [4]).

In mammals, differences in mode of reproduction and developmental timing between placentals (e.g. cats, pangolins, rats and humans) and marsupials (e.g. kangaroos, Tasmanian devils and opossums) have been hypothesized to drive their disparate evolutionary histories [5–9]. Marsupials in general have a unique reproductive mode with an extremely short period of internal gestation of the fetus inside the mother (see [10] for exception). As a result of this accelerated gestation, marsupials give birth to neonates in an embryonic condition, which then need to travel from the birth canal powered by their forelimbs, in order to attach to the mother's teat to suckle [11]. Because of these early functional requirements, the morphology of newly born marsupials is a mosaic of structures that develop early (such as the forelimb [7,8,12,13] and oral apparatus [14,15]) and

late (such as the hind limbs and braincase [8,12,13,16]). Consequently, the functional requirements imposed by this distinctive reproductive mode may have constrained the evolution of marsupials and cause their relatively low diversity and reduced morphological variation (i.e. disparity) in comparison to placentals [5,6]. This hypothesis has been extensively debated in the literature, and previous authors have found contrasting results [7,12,13,16–19]. Furthermore, some studies have argued that other factors may have impacted mammalian evolution, such as historical biogeographic distributions [18]. By contrast to placentals, marsupials and their extinct relatives currently occupy a smaller terrestrial surface with less diverse types of environments and, might have had fewer opportunities to adapt to different ecological contexts [18].

To date, most of the studies investigating the role of developmental constraints on morphological evolution in mammals focus either on marsupials [12,14,15,20], or placentals [21]. A handful of studies have included a broad sample of placental and marsupial mammals [8,16,19], and focused on the limb elements [7,12,16,22,23] or the skull, but none have focused on the morphological evolution of the jaw, the one structure that is uniformly and unambiguously functioning in neonatal marsupials. Whereas some marsupials do not crawl to the mother's teat (and thus do not require functional forelimbs at birth) [24], and the skull has multiple functions in sensory and nervous systems, the jaw has one clear primary purpose, for feeding, and serves this function immediately after birth in all marsupial neonates. More broadly, previous studies of jaw morphology routinely use operational taxonomic unit and diversity scoring [25] or linear data [26], but the marsupial jaw, in particular, involves unusual processes (e.g. the inflected angular) that require a three-dimensional approach.

In this study, we use a broad and diverse sample of modern and fossil eutherian and metatherian mammals (the crown and stem clades of placentals and marsupials, respectively) to test whether functional constraints during development may have limited morphological evolution in the metatherian jaw relative to that of eutherians. To do so, we reconstruct the evolution of jaw shape and quantify the relative influences of reproductive mode and function (diet) on its morphology. If functional constraints experienced during early development have indeed limited the evolution of the jaw in metatherians, we predict that mandible disparity and rate of evolution should be lower and slower in metatherians than in eutherians.

## 2. Material and methods

### (a) Material
Our dataset is composed of the mandible from 151 individuals representing 52 modern and 3 fossil metatherian species and 75 modern and 20 fossil eutherians. This sample was chosen in order to represent a wide phylogenetic breadth, as well as diversity in ecology, morphology and function among terrestrial eutherians and metatherians. One hundred and thirty meshes were generated for this study and 21 were collected from online repositories (electronic supplementary material, table S1). All information about the specimens and the definition of their dietary and reproductive mode came from the literature (see electronic supplementary material, table S1), especially for extinct taxa. Note that it is assumed in this study that all eutherians give birth

to more developed newborns (from altricial to precocial) whereas all metatherians give birth to less-developed newborns (highly altricial). Furthermore, we consider that all metatherians share the reproductive mode observed in extant marsupials, although this is ambiguous for stem taxa (e.g. borhyaenids). Dietary categories were defined as carnivorous, lingual feeder, browser, insectivorous, grazer, tuberivores, frugivorous and omnivorous following the literature [27–29].

### (b) Quantification of mandibular shape using three-dimensional geometric morphometrics
A total of 16 landmarks and 98 curve sliding semilandmarks were identified to comprehensively capture the shape of the mandible (electronic supplementary material, figure S1 and tables S2, S3). All landmarks and curve semilandmarks were taken manually by the same person (A.-C.F.) using Checkpoint (Stratovan, Davis, CA, USA). In order to transform the curve semilandmarks into geometrically homologous points, a three-dimensional sliding semilandmark procedure [30,31] was performed following the protocol described in previous studies [4,32–34]. In order to slide all of the curve semilandmarks while minimizing bending energy, we used the 'slider3d' function from the Morpho R package [35,36]. Finally, a Procrustes superimposition was performed using the 'gpagen' function from the geomorph R package [37] to remove the effects of non-biological variation (rotation, translation and isometric size).

### (c) Phylogenetic tree
A phylogenetic tree was constructed to incorporate all of the species used in this study, based on a recently dated molecular tree for extant mammals [38]. Fossils were grafted onto this tree based on recent morphological studies (see the electronic supplementary material), with tip positions informed by their occurrences.

### (d) Shape variation depending on infraclass and diet
Both non-phylogenetic and phylogenetic principal components analysis (PCA) [39] were used to visualize jaw shape variation across eutherians and metatherians. To do so, we used, respectively, the functions 'gm.prcomp' from the R geomorph package [37], and 'phyl.pca' and 'phylomorphospace' from the R phytools package [39,40]. Next, we tested for shape differences between infraclass and diet using a type II phylogenetic multivariate analysis of variance (MANOVA) in R mvMORPH package [41]. The multivariate phylogenetic linear models were fitted with a Pagel's lambda by penalized likelihood using the 'mvgls' function [41]. Pagel's lambda has the advantage to provide increased flexibility in estimating the error structure and it is equivalent to fitting a phylogenetic mixed model while accounting for departure from Brownian motion (BMM) [41–44]. Subsequently, this model was used as input in the 'manova.gls' function (electronic supplementary material, table S4). The significance of each type II phylogenetic MANOVA was assessed using a Pillai statistic and 1000 permutations. Finally, to identify morphological convergence depending on diet and infraclass, we used the phylogenetic ridge regression method of the RRphylo package in R [45]. We first performed a phylogenetic ridge regression on the tree and with the first 24 PC scores (accounting for 95% of the overall variance) using the 'RRphylo' function in order to obtain the branch-wise evolutionary rates and the ancestral character estimates at nodes. Next, we establish morphological convergence for each diet category using the 'search.conv' function under 'state' cases on PC scores. From these analyses, we retrieve the mean angle (mean angle between species within the dietary category), and the $p$ of the mean angle (significance level for mean angle) [45].

### (e) Disparity based on infraclass

To assess and compare the disparity between eutherian and metatherian reproductive modes, we calculated Procrustes disparity using the 'morphol.disparity' and the 'dispRity.per.group' functions, respectively, in the Geomorph and the DispRity packages in R [37,46]. Procrustes disparity allows to estimate the Procrustes variance per group by using the residuals of a linear model fit. Here, we used as input the Procrustes coordinates in order to calculate the sum of the diagonal elements of the infraclass group covariance matrix divided by the number of observations in the infraclass group. To test for Procrustes disparity difference between infraclass, a pairwise comparison was performed using the 'morphol.disparity' function in the Geomorph package in R [37]. We further used a non-parametric Wilcoxon test to assess the significance of differences in disparity between eutherians and metatherians using the 'wilcox.test' function in the DispRity package in R [46]. Disparity analyses were performed on the entire dataset (extinct and extant species) and on a dataset comprising only modern mammals.

### (f) Rates of morphological evolution depending on infraclass

In order to assess and compare morphological rates of evolution between infraclass, we calculated evolutionary rates for each infraclass category for both entire and extant species datasets. To do so, the reconstructed history between eutherian and metatherian reproductive modes categories on which a state-specific BMM model were fitted was obtained through stochastic character mapping across a sample of 100 trees using the 'make.simmap' function and an 'ARD' model in the R package phytools [39]. Model fit was performed using a state-specific BMM model in the 'mvgls' function in mvMORPH R package v. 1.1.4 [41,47] both using the entire dataset and on the modern dataset only. Finally, we estimated a branch-specific evolutionary rates and rate shifts using the variable rates model implemented in BayesTraits v. 3 (http://www.evolution.rdg.ac.uk/). Shifts in the rate of continuous trait evolution (modelled by a BMM process) were detected using a reversible-jump Markov chain Monte Carlo algorithm. The phylogenetic principal components (PCs) accounting for 95% of the overall variation in jaw shape were used as input (i.e. the first 33 phylogenetic PCs for the entire dataset and the first 32 for the extant dataset). Ten independent chains were run for 200 000 000 iterations, sampling every 10 000 iterations and the first 25 000 000 iterations were discarded as burn-in. Trace plots were examined and two independent chains that were stationary after burn-in were kept. We assessed the effective sample size (ESS) of the posterior samples (ESS > 100) as well as the convergence of the chains, with a Gelman and Rubin's convergence diagnostic, using, respectively, the functions 'effectiveSize' and 'Gelman.diag' implemented in the R coda package (see electronic supplementary material, figures S2, S3 and tables S4–S7). Finally, we plotted the results of the analyses on the tree using the function 'mytreebybranch' (https://github.com/anjgoswami/salamanders/blob/master/mytreerateplotter.R). The branch-specific average rate and the posterior probability of rate shifts were summarized from the posterior samples using the 'rjpp' and 'plotShift' functions in the btrtools R package v. 0.0.0.9000 (https://github.com/hferg/btrtools/tree/master/R).

## 3. Results

### (a) Mandibular morphological variation

The results of the PCA and the phylogenetic PCA (figures 1 and 2; electronic supplementary material, figures S4 and S5) show that metatherians occupy a smaller area of the morphospace in comparison to eutherians. The distribution of metatherian species overlaps largely with that of the eutherians, and species of metatherians and eutherians with similar diets tend to have similar mandibular shapes, as illustrated in figure 2. The first axis accounting for 32.14% of the overall variance differentiates species with an elongated mandibular body and an extremely reduced ramus on the negative part in comparison to those with a short mandibular body and well-developed ramus on the positive side. The second axis (15.06% of the variance) separates species with a strong mandibular articulation positioned high and a reduced coronoid process on the negative side in comparison to those with a mandibular articulation that is positioned low on the jaw and posteriorly oriented, as well as bearing a well-developed coronoid process.

### (b) Morphological difference and convergence depending on diet between eutherians and metatherians

The results of the phylogenetic MANOVA found no mandibular shape differences among metatherian and eutherian reproductive modes for both datasets (entire dataset: Pillai's test = 0.25, $p = 0.1$; extant dataset: Pillai's test = 0.28, $p = 0.16$). By contrast, shape differences between dietary groups were found in both datasets (entire dataset: Pillai's test = 5.86, $p = 0.001$; extant dataset: Pillai's test = 6.25, $p = 0.001$). The results of the convergence analyses performed on each dietary category are significant for the mean angle ($p < 0.05$) in lingual feeders, carnivores, omnivores, browsers, tuberivores, grazers and insectivores (table 1).

### (c) Disparity differences depending on infraclass

There is a significant difference in the disparity of the jaw shape between infraclass (Procrustes variance for the entire dataset: Eutheria = 0.030, Metatheria = 0.016, $p$-value = 0.001; Procrustes variance for extant dataset: Eutheria = 0.032, Metatheria = 0.015, $p$-value = 0.001; electronic supplementary material, figure S6), with eutherians having a higher disparity than metatherians for both datasets (entire dataset and modern species only dataset).

### (d) Evolutionary rates

The rate of evolution of the mandible is significantly different between eutherians and metatherians (entire dataset: $\sigma$Eutheria = $9.470920 \times 10^{-7}$, $\sigma$Metatheria = $3.114322 \times 10^{-7}$, $p$-value = 0.0001; extant dataset: $\sigma$Eutheria = $8.237860 \times 10^{-7}$, $\sigma$Metatheria = $1.471253 \times 10^{-7}$, $p$-value = 0.0001), with metatherians showing a slower rate of evolution than eutherians in both datasets. Branch-specific rate reconstructions in BayesTraits identified major shifts in the rate of mandible evolution occurred early in the evolution of mammals, at the transition between eutherians and metatherians (figure 3; electronic supplementary material for results on extant species, electronic supplementary material, figure S7). A slower rate of evolution in jaw shape is observed in metatherians in comparison to eutherians, with exception of the macropodids which are characterized by an acceleration of the rate of morphological evolution at the base of the clade.

Proc. R. Soc. B 288: 20210319

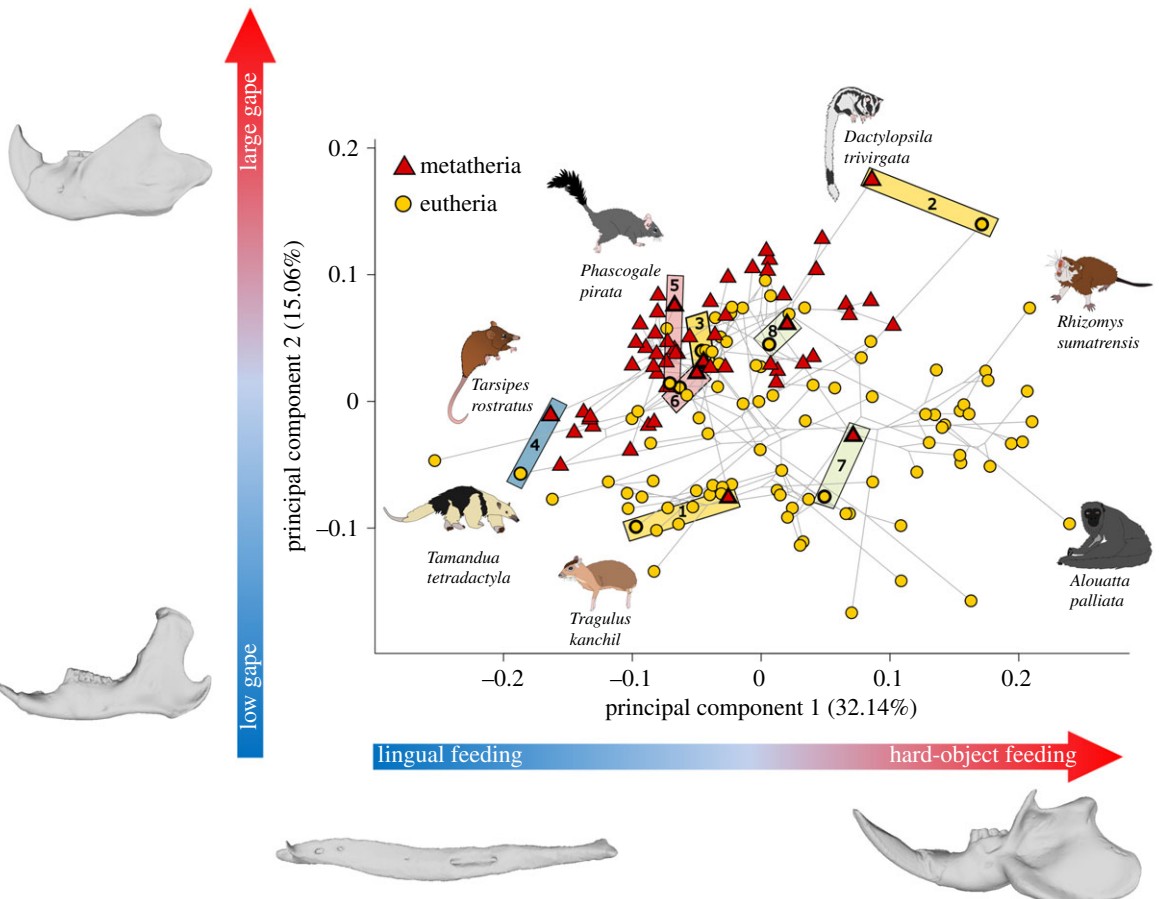

**Figure 1.** Phylomorphospace illustrating the first two principal components of jaw shape across mammals. The phylogeny is mapped onto the morphospace in light grey. Red triangles represent metatherian species and yellow circles represent eutherians. Jaw shapes are depicted at the positive and negative extremes of each axis. Coloured, shaded boxes with numbers represent an example of similar jaw shape due to dietary convergences between metatherian and eutherian species and are further illustrated in figure 2. Box colours are as follows: yellow as insectivorous, light red as carnivorous, blue as lingual feeders and light green as browsers.

## 4. Discussion

Understanding the impacts of heterochronic shifts on evolutionary trajectories is of fundamental interest in organismal and evolutionary biology. Changes in jaw function experienced during ontogeny may impose or release developmental constraints, with consequences for morphological variation and, ultimately, evolutionary diversity. The three subclasses of mammals show major differences in reproductive strategy, from the species-poor, egg-laying monotremes, to marsupials and placentals. Assessing the impact of these reproductive strategies on the divergent evolutionary histories of these clades requires an assessment of shape evolution in a representative sample of modern and fossil species, as conducted in the limb and cranial skeletons [7,22,48–50]. Here, we advance this work with the first study investigating how the marsupial reproductive strategy has impacted the evolution of the jaw, using a diverse sample of modern and fossils species with a range of ecologies for both eutherians and metatherians. Because the jaw has one primary purpose and is uniformly functional immediately after birth in all marsupials, it is arguably one of the best bony elements for assessing the impact of the marsupial reproductive strategy on the evolution of morphological diversity on this clade. Our results unambiguously support the prediction that early suckling and related accelerated development of the jaw constrains its evolution in metatherians. Specifically, our results demonstrate that metatherian mandibles have a lower disparity and slower rate of evolution in comparison to eutherians.

Previous studies of other skeletal systems (limbs and skull) have also largely supported the hypothesis of development constraint in metatherians. Prior investigations of adult skull disparity [48,50] and ontogenetic trajectories of skull shape change [48] demonstrated that metatherians have significantly less disparity and more aligned ontogenetic trajectories than eutherians in the early-ossifying oral region of the skull, but not in the late-ossifying neurocranial region, supporting the hypothesis that suckling function constrains the development and evolution of the oral components of the skull. Several previous studies of postcranial elements (forelimb and hind limb) using adult and ontogenetic data marsupials and placentals found strong support for constraints in metatherian morphological evolution at the macroevolutionary [7] and microevolutionary scales [22,23,51], including comparing ontogenetic trajectories of forelimb shape [7]. Only one quantitative analysis to date [12] has concluded that limb evolution in marsupials may not be constrained by development. However, that study investigated this hypothesis using only marsupials and qualitatively compared their results to those from other studies of placentals, rather than directly comparing data from both clades [12]. Analyses of disparity and rate of evolution are difficult to compare across studies, particularly when there are differences in data types and samples. Consequently, the results obtained for one dataset (e.g. marsupials) in one study may not be directly comparable to those from another study with non-overlapping samples (e.g. placentals)

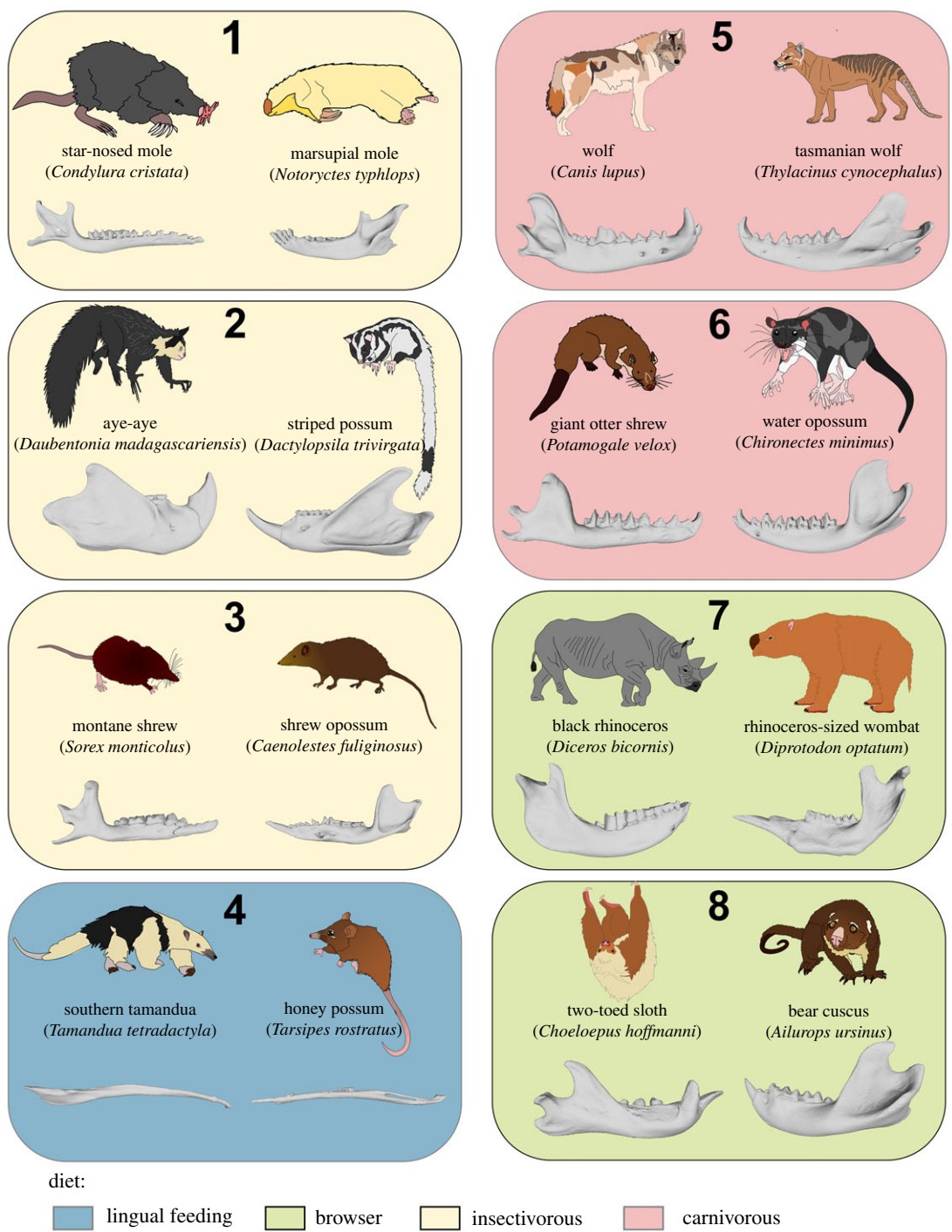

**Figure 2.** Exemplar convergences in jaw shape between species of metatherians and eutherians with similar diets. Box colours and numbers correspond to those indicated in the phylomorphospace in figure 1.

or different morphometric data. In addition, comparisons only within marsupials [12] cannot fully test for developmental constraints due to the marsupial reproductive strategy because they lack an appropriate baseline. To assess whether or not the marsupial developmental strategy constrains the evolution of specific structures that are functionally engaged early in postnatal ontogeny, the appropriate comparison is not between different structures within the marsupial skeleton, which can differ extensively in shape, function and disparity for many reasons, but with disparity of the same structure in taxa that do not have that developmental constraint (i.e. placentals [20]).

Our shape analyses demonstrate that metatherians occupy the same parts of jaw morphospace as eutherians. However,

they highlight that metatherians occupy a smaller area of that morphospace compared to eutherians. Moreover, our results showed that metatherians never invade some parts of the eutherian morphospace, such as Artiodactyla, Rodentia and part of the primate morphospace, even if both groups display the same range of terrestrial ecological diversity. Nonetheless, our pMANOVA analyses did not find any overall differences in jaw shape between metatherians and eutherians, suggesting extensive convergence across both clades between species with similar diets. In support of this result, it is evident in the morphospace that some species with similar diets or similar biomechanical requirements occupy the same parts of the morphospace (figures 1 and 2). For example, the first PC axis differentiates species using lingual feeding (nectarivorous

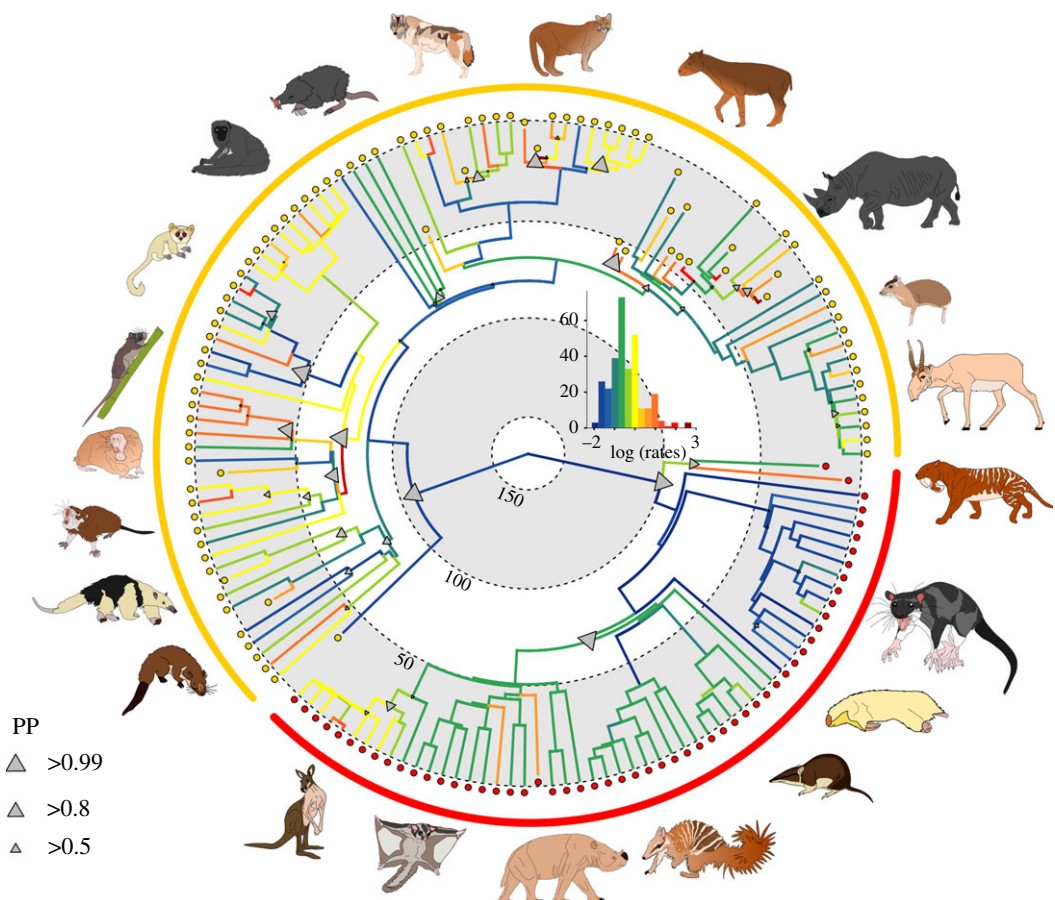

**Figure 3.** Evolutionary rates and rate shifts for mandibular shape in mammals. Branch rates are indicated by the colour gradients with warmer colours corresponding to a faster rate and cooler colours to a slower one. High probability shifts in morphological rates are indicated by grey triangles. The relative size of the triangles represents the posterior probabilities (PP) of rate shifts (see electronic supplementary material, figure S8 for tree with PP and the results for the extant dataset electronic supplementary material figures S7 and S9).

**Table 1.** The results of the convergence performed on the jaw shape depending on diet within mammals. Mean angle corresponds to the mean angle between species within the same diet; and *p* mean angle corresponds to the significance level for mean angle. Values in bold indicate statistically significant results.

|  | mean angle | *p* mean angle |
|---|---|---|
| lingual feeding | 44.72895554 | **0.001** |
| carnivorous | 70.00511249 | **0.001** |
| omnivorous | 85.18603 | **0.007** |
| browser | 86.48755 | **0.035** |
| mixed feeder | 74.44509 | 0.087 |
| tuberivorous | 52.65481 | **0.02** |
| grazer | 77.11483 | **0.001** |
| frugivorous | 85.08764 | 0.064 |
| insectivorous | 83.45912016 | **0.021** |

and myrmecophageous species) from species eating 'hard' or difficult to reduce items (bamboo, seeds and roots). Species eating hard items display a short and robust mandibular body, a ramus with an extremely well-developed angular process and a mandibular condyle with a rounded articulation. The second PC axis differentiates species with jaws optimized for feeding at low gape, with a strong mandibular articulation positioned high and a reduced coronoid process, from species

optimized for feeding at large gapes with a mandibular articulation that is positioned low on the jaw and a well-developed coronoid process. It is interesting to note that the most striking morphological feature used in the literature to distinguish the marsupial jaw from the placental one is the medially inflected angular process. Surprisingly, this morphological feature does not appear to clearly separate marsupials from placentals in the morphospace. Thus, it seems that diet specializations are a more important driver of jaw shape difference in mammals, particularly the difference between species eating soft food in comparison to those eating hard food. Several of these convergences are illustrated in figures 1 and 2, lingual feeders such as termite-eating specialists southern tamandua (*Tamandua tetradactyla*), numbat (*Myrmecobius fasciatus*) and pangolin (*Manis javanica*) fall in the same part of the morphospace as the nectarivorous honey possum (*Tarsipes rostratus*). Jaw shape convergence between metatherians and eutherians are also observed in carnivorous species such as the thylacine (*Thylacine cynocephalus*) and the wolf (*Canis lupus*), and the water opossum (*Chironectes minimus*) and the giant otter shrew (*Potamogale velox*). Similarly, insectivorous adaptations drive similar jaw shapes in species such as the eutherian star-nosed mole (*Condylura cristata*) and the marsupial mole (*Notoryctes typhlops*), the aye-aye (*Daubentonia madagascariensis*) and the striped possum (*Dactylopsila trivirgata*), as well as the montane shrew (*Sorex monticolus*) and the shrew opossum (*Caenolestes fuliginosus*). Other notable examples displaying similar jaw shapes are browsers such as the black rhinoceros (*Diceros bicornis*) and the rhinoceros-sized wombat (*Diprotodon*

*optatum*), as well as the two-toed sloth (*Choloepus hoffmanni*) and the bear cuscus (*Ailurops ursinus*). These results support previous ecomorphological studies on the convergence between marsupials and placentals (e.g. [52,53]). However, they also demonstrate that these convergences occurred only in a small part of the morphospace and in some dietary groups, as demonstrated by the convergence analyses which are significant in lingual feeder, carnivore, omnivore, browser, tuberivore, grazer and insectivore species. Combined, these results suggest that the morphological evolution of the metatherian jaw is developmentally constrained, relative to eutherians. Moreover, it is interesting to note that metatherians and eutherians overlapped and have no significant jaw shape difference as suggested by the results of the PCA and the phylogenetic MANOVA whereas they show a significant difference in disparity and rate of evolution. These results tend to indicate that the shape variation and rate evolution between both mammalian groups seems to be the main driver of their evolutionary difference rather than their actual shape difference itself. Therefore, this study provides new insights concerning the impact of the developmental timing on the pattern of morphological evolution in mammals and confirms that development can be a major driver or constraint of morphological evolution across major lineages, even within clades that are entirely viviparous.

## 5. Conclusion

Our research demonstrates that developmental timing and ontogenetic function have had a significant influence on jaw shape evolution in mammals, with lower disparity and a slower rate of evolution observed in metatherians compared to eutherians. It also shows that metatherians occupy a smaller area of the jaw morphospace than eutherians do, despite similar terrestrial ecological diversities. Jaw shape does not differ significantly between eutherians and metatherians either, reflecting significant convergences driven by similar diets. These results suggest that functional constraints associated with different ecologies may impact on mandibular shape in mammals, but to a lesser degree than developmental timing.

Ethics. Our study did not include human subjects, live animals or fieldwork.

Data accessibility. All analytical code is freely available in R packages and other open software. mvMORPH v. 1.1.4 is available from github.com/JClavel/mvMORPH [install_github ('JClavel/mvMORPH', ref='devel_1.0.4')]. All data are available as electronic supplementary material, information and from the Dryad Digital Repository: https://doi.org/10.5061/dryad.b8gtht7c5 [54]. The mesh data that support the findings of this study will be deposited in the Phenome10 K repository (http://phenome10k.org/) pending acceptance of the paper or are already available on MorphoSource, DigiMorph and Sketchfab.

Authors' contributions. A.-C.F. and A.G. designed the study. A.-C.F., R.P.M., E.N., V.F. and A.G. acquired and processed the CT data. A.-C.F acquired the geometric morphometric data. A.G. constructed the phylogenetic framework. A.-C.F. constructed the shape analyses. A.-C.F. and A.G. wrote the initial draft of the manuscript. A.-C.F., C.D., R.P.M., E.N., V.F. and A.G. contributed to the interpretation of the data and to the editing of subsequent drafts of the manuscript.

Competing interests. The authors declare no competing interests.

Funding. This work was funded by the European Research Council (grant no. STG-2014-637171 to A.G.). CT data from DigiMorph.org on the NSF grant nos. EF-0334952, IIS-9874781 and IIS-0208675. CT data from Morphosource on: the European Research Council (ERC) starting grant TEMPO (ERC-2015-STG-677774 to Roger Benson); the NSF GRFP, DDIG; NSF DEB-1257572 (to Z.J. Tseng); NSF DBI-1701714; NSF DBI-1701769; NSF BCS 1317525 (to D. M. Boyer and E. R. Seiffert), 1552848 (to D. M. Boyer) and Duke University Trinity College of Arts and Sciences; NSF-1701420; Digitization TCN: Collaborative Research: oVert: Open Exploration of Vertebrate Diversity in three dimensions.

Acknowledgements. We would like to thank Dr J. Hutchinson, the associated editor as well as Dr Vera Weisbecker and one anonymous reviewer for their positive and constructive review on our manuscript. We also thank B. Clark for providing training for X-ray CT scanning at the NHM and J. Maisano for giving access to CT data from DigiMorph.org. We thank B. Poole and J. Clavel for providing help and feedback during analyses while using Bayestrait and mvMORPH. We thank J. Van Zoelen for providing access to the three-dimensional mesh of a complete *Diprotodon* mandible. All the following specimens were downloaded from www.MorphoSource.org, Duke University: R. Benson and C. Thompson for providing access to the CT data of *Tarsipes rostratus*; Z. J. Tseng, C. Grohé and J. J. Flynn provided access to Canis lupus data originally appearing in Tseng ZJ, Grohé C, Flynn JJ. 2016. A unique feeding strategy of the extinct marine mammal Kolponomos: convergence on sabretooths and sea otters. *Proceedings of the Royal Society B: Biological Sciences*, the collection of which was funded by the NSF and the American Museum of Natural History Frick Postdoctoral Fellowships; Yale Peabody Museum provided access to these data provided access to *Condylura cristata*, the collection of which was funded by oVert TCN; D. Boyer provided access to the CT data *Cynocephalus*; Idaho Museum of Natural History provided access to *Procyon lotor*, the collection of which was funded by Rick Carron Foundation. Texas A&M AgriLife Research provided access to *Solenodon paradoxus*.

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
