## [Peer Review File · Proceedings of the Royal Society B: Biological Sciences]

Review History

RSPB-2021-0319.R0 (Original submission)

Review form: Reviewer 1

Recommendation

Accept with minor revision (please list in comments)

Scientific importance: Is the manuscript an original and important contribution to its field?

Good

General interest: Is the paper of sufficient general interest?

Good

Quality of the paper: Is the overall quality of the paper suitable?

Excellent

Is the length of the paper justified?

Yes

Should the paper be seen by a specialist statistical reviewer?

Yes

Do you have any concerns about statistical analyses in this paper? If so, please specify them explicitly in your report.

No

It is a condition of publication that authors make their supporting data, code and materials available - either as supplementary material or hosted in an external repository. Please rate, if applicable, the supporting data on the following criteria.

Is it accessible?

Yes

Is it clear?

Yes

Is it adequate?

Yes

Do you have any ethical concerns with this paper?

No

Comments to the Author

Unlike eutherian (placental) mammals, metatherian (marsupial) mammals are born before the end of embryonic development and complete their ontogeny outside of the body. This results in adaptation to the craniofacial skeleton and jaw to ensure feeding is possible at birth. This interesting article investigates the unique life history of marsupials results in developmental constraints limiting the potential morphologies of the lower jaw bone.

I find the data and arguments made clear and convincing, and the paper well written. Therefore I have no major revision requests.

Minor comments:

The diets corresponding to the box colours should be re-stated in figure 2 for ease of reading. This can be either in the legend or, preferably, in the figure itself.

One striking feature of the marsupial jaw compared to that of eutherians is the medially inflected angular process. Whilst mentioned in passing, I am surprised that this feature does not appear to alter the position and the morphospace more, particularly for hard food specialist feeders. perhaps a sentence or two could be added to discuss this.

Review form: Reviewer 2 (Vera Weisbecker)

Recommendation

Accept with minor revision (please list in comments)

Scientific importance: Is the manuscript an original and important contribution to its field?

Excellent

General interest: Is the paper of sufficient general interest?

Excellent

Quality of the paper: Is the overall quality of the paper suitable?

Excellent

Is the length of the paper justified?

No

Should the paper be seen by a specialist statistical reviewer?

No

Do you have any concerns about statistical analyses in this paper? If so, please specify them explicitly in your report.

No

It is a condition of publication that authors make their supporting data, code and materials available - either as supplementary material or hosted in an external repository. Please rate, if applicable, the supporting data on the following criteria.

Is it accessible?

Yes

Is it clear?

Yes

Is it adequate?

Yes

Do you have any ethical concerns with this paper?

No

Comments to the Author

This study uses a sophisticated pipeline of geometric morphometric and phylogenetic methods to provide the best-supported comparative investigation of the claim that marsupial mammals have a developmentally constrained jaw and are therefore potentially less “adaptable”. The paper is very well written and clear, with excellent analyses, so that I have very few comments and no substantial issues at all. I love the availability of extensive code scripts!

The introduction is simple, clear, well-argued, and well-referenced. I have no suggestions for improvements.

The base methods are mostly following well-established protocols, but the phylogenetic methods are much more sophisticated than what is usually done, e.g. phylogenetic linear models that not just depend on an assumption of Brownian motion, and the excellent phylogenetic ridge regression for convergence. The BayesTraits pipeline is wonderful, it is so sad that this can't be implemented in R (yet).

A minor method/results clarity issue – at first I was looking for a reproductive mode variable in Table S1, but in the code “reproductive mode” seems to refer to the infraclass variable (i.e. metatherians vs. eutherians), rather than an actual reproductive mode – I also later found this in the text but it could be easier to understand if “reproductive mode” was replaced with “infraclass”, and perhaps an explanation that the main distinction assumed is between birth maturities.

The discussion is also clear and comprehensive.

The roughly page-long passage that raises validity concerns with Martin-Serra and Benson's paper might need some shortening, it seems out of scope because it deals with broader issues of serial homology and differential constraints in limbs, which this study is not addressing.

I thought a bit about the differences between the pMANOVA and the PCA morphospace. It sounds like the pMANOVA is done on the whole variation in the dataset whereas the PCA "only" accounts for ~half of the variation. Also, marsupials overlap with placentals on both PCs (but particularly on the much more important PC1). So this suggests that the differences in disparity and evolutionary rates between marsupials and placentals are a much greater point of difference than the actual differences in shape that we see on PC1/2 space. I find that fascinating and perhaps it could be emphasized, but this is just a suggestion.

P. 4, line 36: comma after "gestation"

P. 4, line 46: comma after "evolution"

p. 5, line 60: no comma after "jaw"

P. 8, line 124: It is called a Wilcoxon test (the R function is called "wilcox.test" though).

p. 8, the Procrustes disparity could be explained more clearly as to what the disparity measures (because there are different ways of measuring disparity) .

p. 191 it should probably be "with exception of the macropodids"

P. 14, line 249 – primate (not primates)

P. 17, Acknowledgements – it is *Tarsipes rostratus* (not *rostratum*).

I would recommend using a more precise term than "functional pressures", which sounds a bit like evolutionary selection pressure but is here used in the context of the evolution of individual development. Maybe "jaw function"?

Vera Weisbecker

Decision letter (RSPB-2021-0319.R0)

18-Mar-2021

Dear Dr Fabre:

Your manuscript has now been peer reviewed and the reviews have been assessed by an Associate Editor. The reviewers' comments (not including confidential comments to the Editor) and the comments from the Associate Editor are included at the end of this email for your reference. As you will see, the reviewers and the Editors have raised some concerns with your manuscript and we would like to invite you to revise your manuscript to address them.

Research ethics:

Use of animals and field studies:

It is a condition of publication that you make available the data and research materials supporting the results in the article. Please see our Data Sharing Policies (<https://royalsociety.org/journals/authors/author-guidelines/#data>). Datasets should be deposited in an appropriate publicly available repository and details of the associated accession number, link or DOI to the datasets must be included in the Data Accessibility section of the article (<https://royalsociety.org/journals/ethics-policies/data-sharing-mining/>). Reference(s) to datasets should also be included in the reference list of the article with DOIs (where available).

If you wish to submit your data to Dryad (<http://datadryad.org/>) and have not already done so you can submit your data via this link [http://datadryad.org/submit?journalID=RSPB&manu=\(Document not available\)](http://datadryad.org/submit?journalID=RSPB&manu=(Document%20not%20available)), which will take you to your unique entry in the Dryad repository.

Please submit a copy of your revised paper within three weeks. If we do not hear from you within this time your manuscript will be rejected. If you are unable to meet this deadline please let us know as soon as possible, as we may be able to grant a short extension.

Best wishes,
Dr John Hutchinson, Editor
mailto:proceedingsb@royalsociety.org

Associate Editor
Comments to Author:

We have now received two thoughtful reviews of this manuscript. I am delighted to share the news that both reviewers found the manuscript interesting and well assembled. Both reviewers have provided a few simple recommendations/queries, and I suggest the authors address all of these. I agree with the reviewers that this is an excellent study and manuscript.

Reviewer(s)' Comments to Author:

Referee: 1

Comments to the Author(s)

Unlike eutherian (placental) mammals, metatherian (marsupial) mammals are born before the end of embryonic development and complete their ontogeny outside of the body. This results in adaptation to the craniofacial skeleton and jaw to ensure feeding is possible at birth. This interesting article investigates is the unique life history of marsupials results in developmental constraints limiting the potential morphologies of the lower jaw bone.

I find the data and arguments made clear and convincing, and the paper well written. Therefore I have no major revision requests.

Minor comments:

The diets corresponding to the box colours should be re-stated in figure 2 for ease of reading. This can be either in the legend or, preferably, in the figure itself.

One striking feature of the marsupial jaw compared to that of eutherians is the medially inflected angular process. Whilst mentioned in passing, I am surprised that this feature does not appear to alter the position and the morphospace more, particularly for hard food specialist feeders. perhaps a sentence or two could be added to discuss this.

Referee: 2

Comments to the Author(s)

This study uses a sophisticated pipeline of geometric morphometric and phylogenetic methods to provide the best-supported comparative investigation of the claim that marsupial mammals have a developmentally constrained jaw and are therefore potentially less "adaptable". The paper is very well written and clear, with excellent analyses, so that I have very few comments and no substantial issues at all. I love the availability of extensive code scripts!

The introduction is simple, clear, well-argued, and well-referenced. I have no suggestions for improvements.

The base methods are mostly following well-established protocols, but the phylogenetic methods are much more sophisticated than what is usually done, e.g. phylogenetic linear models that not just depend on an assumption of Brownian motion, and the excellent phylogenetic ridge regression for convergence. The BayesTraits pipeline is wonderful, it is so sad that this can't be implemented in R (yet).

A minor method/results clarity issue – at first I was looking for a reproductive mode variable in Table S1, but in the code “reproductive mode” seems to refer to the infraclass variable (i.e. metatherians vs. eutherians), rather than an actual reproductive mode – I also later found this in the text but it could be easier to understand if “reproductive mode” was replaced with “infraclass”, and perhaps an explanation that the main distinction assumed is between birth maturities.

The discussion is also clear and comprehensive.

The roughly page-long passage that raises validity concerns with Martin-Serra and Benson’s paper might need some shortening, it seems out of scope because it deals with broader issues of serial homology and differential constraints in limbs, which this study is not addressing.

I thought a bit about the differences between the pMANOVA and the PCA morphospace. It sounds like the pMANOVA is done on the whole variation in the dataset whereas the PCA “only” accounts for ~half of the variation. Also, marsupials overlap with placentals on both PCs (but particularly on the much more important PC1). So this suggests that the differences in disparity and evolutionary rates between marsupials and placentals are a much greater point of difference than the actual differences in shape that we see on PC1/2 space. I find that fascinating and perhaps it could be emphasized, but this is just a suggestion.

P. 4, line 36: comma after “gestation”

P. 4, line 46: comma after “evolution”

p. 5, line 60: no comma after “jaw”

P. 8, line 124: It is called a Wilcoxon test (the R function is called “wilcox.test” though).

p. 8, the Procrustes disparity could be explained more clearly as to what the disparity measures (because there are different ways of measuring disparity) .

p. 191 it should probably be “with exception of the macropodids”

P. 14, line 249 – primate (not primates)

P. 17, Acknowledgements – it is *Tarsipes rostratus* (not *rostratum*).

I would recommend using a more precise term than “functional pressures”, which sounds a bit like evolutionary selection pressure but is here used in the context of the evolution of individual development. Maybe “jaw function”?

Vera Weisbecker

Author's Response to Decision Letter for (RSPB-2021-0319.R0)

See Appendix A.

Decision letter (RSPB-2021-0319.R1)

29-Mar-2021

Dear Dr Fabre

I am pleased to inform you that your manuscript entitled "Functional constraints during development limit jaw shape evolution in marsupials" has been accepted for publication in Proceedings B. No further revisions are required. Congratulations!

Data Accessibility section

Open Access

Your article has been estimated as being 9 pages long. Our Production Office will be able to confirm the exact length at proof stage.

Paper charges

Sincerely,

Dr John Hutchinson

Appendix A

Zurich, the 20th of March 2021,

Dear Dr Hutchinson,

We are pleased to submit our revised manuscript “Functional constraints during development limit jaw shape evolution in marsupials” (RSPB-2021-0319) by Anne-Claire Fabre, Carys Dowling, Roberto Portela Miguez, Vincent Fernandez, Eve Noirault and Anjali Goswami for publication in *Proceedings of the Royal Society B*. We would like to thank both reviewers, the associated editor and yourself for the positive and constructive review of our paper, and we feel that the comments and suggestions provided have greatly strengthened this work. Among the major changes, we re-did figure 2 by adding the colour categories as a legend. We have included the rewording and all the supplementary information required in the comments and suggestions of both reviewers. We would like to thank you once again for the positive and constructive reviews and hope that our paper is now suitable for publication in the *Proceedings of the Royal Society B*. Below you will find a detailed point-by-point reply to the comments of both referees.

Please feel free to contact us if there are any questions concerning this manuscript submission.

Sincerely,

Dr. Anne-Claire Fabre

Associate Editor

Comments to Author:

We have now received two thoughtful reviews of this manuscript. I am delighted to share the news that both reviewers found the manuscript interesting and well assembled. Both reviewers have provided a few simple recommendations/queries, and I suggest the authors address all of these. I agree with the reviewers that this is an excellent study and manuscript.

OUR REPLY: We thank the associated editor for this positive review and we have modified the manuscript in the light of the comments of both reviewers

Reviewer(s)' Comments to Author:

Referee: 1

Comments to the Author(s)

Unlike eutherian (placental) mammals, metatherian (marsupial) mammals are born before the end of embryonic development and complete their ontogeny outside of the body. This results in adaptation to the craniofacial skeleton and jaw to ensure feeding is possible at birth. This interesting article investigates the unique life history of marsupials results in developmental constraints limiting the potential morphologies of the lower jaw bone.

I find the data and arguments made clear and convincing, and the paper well written. Therefore I have no major revision requests.

OUR REPLY: we thank the reviewer this really positive and constructive review.

Minor comments:

The diets corresponding to the box colours should be re-stated in figure 2 for ease of reading. This can be either in the legend or, preferably, in the figure itself.

OUR REPLY: we agree with the reviewer that the diet corresponding to the box colours was missing and we now added a legend to the figure.

One striking feature of the marsupial jaw compared to that of eutherians is the medially inflected angular process. Whilst mentioned in passing, I am surprised that this feature does not appear to alter the position and the morphospace more, particularly for hard food specialist feeders. perhaps a sentence or two could be added to discuss this.

OUR REPLY: thanks for this really good suggestion, we added one sentence to mention this striking result in the discussion

Referee: 2

Comments to the Author(s)

This study uses a sophisticated pipeline of geometric morphometric and phylogenetic methods to

provide the best-supported comparative investigation of the claim that marsupial mammals have a developmentally constrained jaw and are therefore potentially less “adaptable”. The paper is very well written and clear, with excellent analyses, so that I have very few comments and no substantial issues at all. I love the availability of extensive code scripts!

The introduction is simple, clear, well-argued, and well-referenced. I have no suggestions for improvements.

The base methods are mostly following well-established protocols, but the phylogenetic methods are much more sophisticated than what is usually done, e.g. phylogenetic linear models that not just depend on an assumption of Brownian motion, and the excellent phylogenetic ridge regression for convergence. The BayesTraits pipeline is wonderful, it is so sad that this can’t be implemented in R (yet).

OUR REPLY: we thank the reviewer for her really positive comment about our paper, and we are glad that she enjoyed our study as well as the availability of all the scripts and data. We think that nowadays, it is really important to share methodological pipelines as well as data sets with the scientific community.

A minor method/results clarity issue – at first I was looking for a reproductive mode variable in Table S1, but in the code “reproductive mode” seems to refer to the infraclass variable (i.e. metatherians vs. eutherians), rather than an actual reproductive mode – I also later found this in the text but it could be easier to understand if “reproductive mode” was replaced with “infraclass”, and perhaps an explanation that the main distinction assumed is between birth maturities.

OUR REPLY: we agree that this point needs to be clarified in the manuscript, we added one sentence in the main manuscript as well as in the supplementary information stating:

“Note that it is assumed in this study that all eutherians give birth to more developed newborns (from altricial to precocial) whereas all metatherians give birth to less developed newborns (highly altricial).”

We also used carefully checked the main manuscript and supplementary material in order to use the word “infraclass” when it was more appropriate to use than “reproductive mode”.

The discussion is also clear and comprehensive.

The roughly page-long passage that raises validity concerns with Martin-Serra and Benson’s paper might need some shortening, it seems out of scope because it deals with broader issues of serial homology and differential constraints in limbs, which this study is not addressing.

OUR REPLY: we agree and shortened this paragraph.

I thought a bit about the differences between the pMANOVA and the PCA morphospace. It sounds like the pMANOVA is done on the whole variation in the dataset whereas the PCA “only” accounts for ~half of the variation. Also, marsupials overlap with placentals on both PCs (but particularly on the much more important PC1). So this suggests that the differences in disparity and evolutionary rates between marsupials and placentals are a much greater point of difference than the actual differences in shape that we see on PC1/2 space. I find that fascinating and perhaps it could be emphasized, but this is just a suggestion.

OUR REPLY: thanks for this pertinent suggestion! We added a sentence to the discussion to highlight it!

P. 4, line 36: comma after “gestation”

OUR REPLY: thanks, we changed it

P. 4, line 46: comma after “evolution”

OUR REPLY: thanks, we changed it

p. 5, line 60: no comma after “jaw”

OUR REPLY: thanks, we changed it

P. 8, line 124: It is called a Wilcoxon test (the R function is called “wilcox.test” though).

OUR REPLY: thanks, we changed it

p. 8, the Procrustes disparity could be explained more clearly as to what the disparity measures (because there are different ways of measuring disparity).

OUR REPLY: thanks, we changed it

p. 191 it should probably be “with exception of the macropodids”

OUR REPLY: thanks, we changed it

P. 14, line 249 – primate (not primates)

OUR REPLY: thanks, we changed it

P. 17, Acknowledgements – it is Tarsipes rostratus (not rostratum).

OUR REPLY: thanks, we changed it.

I would recommend using a more precise term than “functional pressures”, which sounds a bit like evolutionary selection pressure but is here used in the context of the evolution of individual development. Maybe “jaw function”?

OUR REPLY: we agree, we used jaw function as suggested.

Vera Weisbecker